# Homochirality through Photon-Induced Denaturing of RNA/DNA at the Origin of Life

**DOI:** 10.3390/life8020021

**Published:** 2018-06-06

**Authors:** Karo Michaelian

**Affiliations:** Department of Nuclear Physics and Application of Radiations, Instítuto de Física, Universidad Nacional Autónoma de México, A. P. 20-364, México, D.F. 01000, Mexico; karo@fisica.unam.mx; Tel.: +52-55-622-5165

**Keywords:** homochirality, origin of life, non-equilibrium thermodynamics, photon dissipation, RNA, DNA, tryptophan

## Abstract

Since a racemic mixture of chiral nucleotides frustrates the enzymeless extension of RNA and DNA, the origin of homochirality must be intimately connected with the origin of life. Homochirality theories have elected to presume abiotic mechanisms for prebiotic enantiomer enrichment and post amplification, but none, so far, has been generally accepted. Here I present a novel hypothesis for the procurement of homochirality from an asymmetry in right- over left-circularly polarized photon-induced denaturing of RNA and DNA at the Archean ocean surface as temperatures descended below that of RNA and DNA melting. This asymmetry is attributed to the small excess of right-handed circularly polarized submarine light during the afternoon, when surface water temperatures were highest and thus most conducive to photon-induced denaturing, and to a negative circular dichroism band extending from 230 to 270 nm for small oligos of RNA and DNA. Because D-nucleic acids have greater affinity for L-tryptophan due to stereochemistry, and because D-RNA/DNA+L-tryptophan complexes have an increased negative circular dichroism band between 230 and 270 nm, the homochirality of tryptophan can also be explained by this hypothesis. A numerical model is presented, demonstrating the efficacy of such a mechanism in procuring homochirality of RNA or DNA from an original racemic solution in as little as 270 Archean years.

## 1. Introduction

Molecules with no plane of symmetry come in two distinct geometrical, but energy-degenerate, forms or mirror images called “enantiomers,” which are labeled as being right (D)- or left (L)-handed depending on their preference for absorption of, respectively, right-handed or left-handed circularly polarized light around their absorption maximum. Chirality in many biological molecules is a result of the tetravalent nature of carbon atoms, often associated with a so-called “alpha-carbon atom” that attaches to a functional group. Energy degeneracy implies that enantiomers have essentially equal formation and degradation probability under near equilibrium conditions (except perhaps for one part in 10^17^ due to the parity non-conserving weak force). However, life today has an overwhelming preference for only one enantiomer; thus, non-equilibrium biochemical reactions must be chirality biased. RNA, DNA, ribose, and deoxyribose are right-handed, while the amino acids of life are left-handed.

Today, incorporation of only the correct enantiomer of the nucelotides into RNA or DNA is guaranteed by an unfailing enzymatic chiral selection process. Such enzymes, however, could not have been available at the very beginnings of life. Without enzyme selection, RNA or DNA extension is severely adversely affected by a racemic (equal concentration of both enantiomers) solution of nucleotides, principally because incorporated nucelotides of the wrong chirality act as extension terminators [1]. Orgel [2] has suggested that this frustration of the copying of polynucleotides is one of the greatest obstacles to understanding the origin of life. A detailed analysis of the complexities involved in the polymerization problem has been given by Avetisov and Goldanskii [3].

In the following section, I briefly review mechanisms proposed for enantiomer enrichment. Equally briefly, I describe the problems associated with the efficacy of each mechanism. In Section 3 and Section 4, I describe how homochirality could have arisen gradually within the first replicating molecules without the need to invoke a prebiotic enantiomer excess or catalytic amplification but instead due to an asymmetry in the UVC photon-induced denaturation of RNA or DNA, which fits consistently within the framework of the thermodynamic dissipation theory of the origin of life [4,5,6]. Section 5 presents a simple model demonstrating the efficacy of the proposed mechanism and explains how D-DNA preference for L-tryptophan could have been selected for through the same mechanism. Conclusions are given in Section 6.

## 2. Prevalent Homochirality Theories

The following is a brief introduction to only the most prevalent theories of homochirality, including a discussion of the difficulties encountered. For more comprehensive reviews, see [7,8].

Most theories for homochirality propose a prebiotic enantiomer excess of the biological molecules, produced by abiotic mechanisms operating either at the Earth’s surface or in space. Potential mechanisms for generating this excess are circularly polarized light either photolysing, photocatalyzing, or photoreacting with the molecules, inorganic chiral clay or crystalline template selectivity, magnetochirality, and the parity violating weak interaction.

Terrestrial circularly polarized light can be generated by distinct mechanisms [9,10]. In order of importance, these are as follows: (1) Sunlight scattered at depth in water becomes linearly polarized. If this light is then totally internally reflected at the water–air interface, the vertical component of its electric field undergoes a phase shift. As observed from below, near the surface, one sees up to 10% circularly polarized light outside Snell’s window [9,11]. (2) Molecular (Rayleigh) atmospheric scattering produces linearly polarized light, and a subsequent aerosol (Mie) scattering (particle size >>λ) gives a maximum circular polarization of up to 0.5% at twilight [12,13]. (3) The intrinsic circular polarization of sunlight itself, about one part in 10^6^ [14]. (4) Sunlight interacting with the Earth’s magnetic field gives circular polarized light through the Faraday effect with an anisotropy factor of 10^−10^ [10].

The maximum optical purity (enantiomer excess) that could be obtained through photocatalyzing or photoreacting is given by g/2 with the anisotropy factor g=Δϵ/ϵ, where the circular dichroism $\Delta \epsilon =\epsilon_{L}-\epsilon_{R}$ and $\epsilon_{L}$ and $\epsilon_{R}$ are the molar absorption coefficients for left- and right-circularly polarized light, and $\epsilon =(\epsilon_{L}+\epsilon_{R})/2$ is the average of these. Empirical studies suggest that the values of g/2 for many different photoreactions are such as to result in optical purity of usually less than 1% [15]. It is therefore improbable that circularly polarized light could have given rise to homochirality on Earth through photocatalyzing or photoreactions without some kind of post-amplification.

On the other hand, optical purity is not limited to g/2 for photolysing, so this has been the mechanism most studied. However, there are three basic difficulties with enantiomer enhancement on Earth through photolysing. First, because the circular polarization of sunlight is small, and since the differential left- compared to right-handed photolysing efficacy is small, a very large amount of molecular material would have to be destroyed in order to obtain 100% chirality. Experiments with camphor, for example, suggest that 20% chirality can be achieved by photolysing 99% of the original racemic material [16]. Homochirality would thus require essentially the complete destruction of the original material. Secondly, averaged over the full diurnal cycle, the net circular polarization of sunlight is zero. Finally, high temperature, metal ions, radiation, and ultraviolet light itself all have the tendency to cause racemization, and this effect is enhanced if the molecules are in water [17].

The smallness of the terrestrial circular polarization of sunlight, and its averaging to zero over the diurnal cycle, has led to the consideration of an extraterrestrial origin of the fundamental molecules of life and their chirality. Astronomical sources, giving potentially much greater circular polarization and intensity than the sun, such as synchrotron radiation from neutron stars with large magnetic fields [15,18], have been proposed. However, very few such circularly polarized light sources have been found to date, and all, so far, have only been identified in the infrared region (albeit, presumably because shorter-wavelength light does not penetrate the extensive dust clouds of space). Furthermore, the synchrotron radiation from these sources is generally white, and non-trivial frequency-dependent dispersion properties of the organic molecules means that the circular dichroism (differential absorption of left- over right-handed circularly polarized light) is both positive and negative in different regions of the spectrum. In fact, integrated over the whole spectrum, circular dichroism sums to zero—the “Kuhn-Condon zero sum rule” (see Section 4 and [19]). Therefore, a net enhancement of either chirality could only be entertained if additional arguments restricting the extraterrestrial light to a relevant frequency range could be found [20].

Finally, it is known that gamma rays, high energy particles, unpolarized UV light itself, and the heat of meteoritic entry into the atmosphere all cause racemization [21,22], so further mechanisms would have to be identified which could keep the molecules in their chiral state during their trip to Earth’s surface. Notwithstanding these difficulties, however, up to 15% enantiomer excess has been claimed for some non-biological amino acids delivered to the Earth in carbonaceous chondrite meteorites such as Murchinson. Biological amino acids found in these meteorites have little, if any, enantiomer excess [23].

Inorganic elements with a preferred chirality could have acted as templates for generating the chirality bias of the molecules of life. Bonner et al. [24,25] found that amino acids are enantioselectively adsorbed on chiral, enantiopure quartz crystals. For example, D-Alanine is bound selectively to D-quartz with an enantiomer excess of up to 20%. Results of several groups claiming to have found a selective adsorption of amino acids on the surfaces of achiral clays have been controversial [24,25]. Although there is clear evidence of a small chiral selectivity by clay minerals, it has been argued that such a small effect may be due to previous absorption of optically active biomolecules produced by living organisms [26]. It is also still unclear why prebiotic minerals or clays could have had an important local chiral bias.

Illumination of a racemic mixture of chiral molecules in a magnetic field with non-polarized light induces an enantiomer excess through the Faraday effect [27,28]. This so-called “magneto-chiral dichroism” is operative on Earth, generating circularly polarized light from the interaction of unpolarized sunlight with the terrestrial magnetic field. However, the anisotropy factor is extremely small, of order 10^−10^ [10]. A further problem is that that the magneto-chiral dichroism effect has an opposite sign on opposite sides of the equator, and that terrestrial magnetic field reversals occur periodically. Very young stars have large magnetic fields due to high rotation rates and are also sources of intense UV light. Such an astronomical magneto-chiral effect would be larger but still small, giving rise to an enantiomer excess of only about 10^−6^ [10].

The weak force is parity violating, breaking the energy degeneracy of the right- and left-handed enantiomers. This was first proposed to be the source of biomolecular homochirality by Ulbricht in 1959 [29]. However, a comparison of the weak energy to thermal energy at the Earth’s surface gives $\Delta E/k_{B}T\approx 10^{-17}$ [18], much too small to be a plausible solution in itself to homochirality. Vester et al. [30] proposed a somewhat different mechanism for an enantioselective reaction originating from the parity violating weak interaction. According to the Vester–Ulbricht hypothesis, the longitudinally polarized β-decay electrons would, when decelerated in matter, lead to circularly polarized bremsstrahlung photons, promoting enantioselective reactions. However, as mentioned above, enantiomer excess is limited to g/2, which, for most relevant reactions, is very small.

In summary, although many mechanisms can be conceived which could have given rise to a small enantiomer excess locally, these alone would not have been sufficient to lead to the homochirality of life. An additional auto-catalytic amplification mechanism [31], or far-from-equilibrium condition [32,33] would have been needed to bring the effect to the level of homochirality. Amplification mechanisms rely on different barrier heights in chemical reactions involving chiral catalysts of a small enantiomer excess. However, in true thermodynamic equilibrium, the products must necessarily be racemic, independently of barrier heights, but if the reaction is incomplete, or driven out of equilibrium, then one of the product enantiomers could be produced, at least in the short term, in much greater quantity than the other [7].

Other far-from-equilibrium theories rely on spontaneous symmetry breaking, a type of second-order phase transition involving a control parameter which passes through a critical value. Spontaneous symmetry breaking through amplification of a microscopic fluctuation in non-equilibrium systems with non-linear kinetic laws has been demonstrated by Prigogine [34].

Amplification, by whatever mechanism, therefore presumes a non-equilibrium situation. Indeed, since life is an out-of-equilibrium phenomena, it is not surprising that many of life’s enzyme-promoted chemical reactions are chirality biased. Although such non-equilibrium ideas for homochirality have been argued to apply in general, and although there exists experimental evidence validating the idea of certain out-of-equilibrium chemical reactions (see [7] and references therein), there has not yet been any demonstration of the principle involving the putative original molecules of life: the nucleic acids—RNA and DNA—or the amino acids.

The following section describes a novel non-equilibrium thermodynamic solution to the homochirality problem in which the asymmetry arises as a natural part of the early RNA or DNA enzymeless replication process, which is postulated to be driven by an externally imposed generalized force acting over the system (the Archean solar photon potential) aided by other environmental factors, principally high surface temperatures.

## 3. Photon-Induced Melting of RNA and DNA

The Earth’s surface during the Archean (3.8–2.5 Ga) was subjected to intense ultraviolet light within the 210–290 nm wavelength region [35,36]—the result of a young Sun [37] and the lack of UV absorbing oxygen and ozone in the Earth’s atmosphere. RNA and DNA are extraordinary absorbers and dissipators of UV light within this spectral region [38]. According to the thermodynamic dissipation theory of the origin of life [4,5,6], life arose as microscopic dissipative molecular structuring [39] under the Archean photon flux, not only to dissipate this potential directly but also indirectly through the molecules acting as catalysts for the water cycle by transforming absorbed light into heat, thereby augmenting evaporation at the ocean surface. Such a scenario connects the visible light dissipation by plants and cyanobacteria today with UVC light dissipation by RNA, DNA, and other fundamental molecules in the Archean [40], and emphasizes life’s enduring involvement with the water cycle [41].

Geochemical evidence in the form of ^18^O/^16^O ratios found in cherts of the Barberton greenstone belt of South Africa point to an Earth’s surface temperature of around 80 °C at 3.8 Ga [42] and 70 ± 15 °C during the 3.5–3.2 Ga era [43]. These temperatures, near the beginnings of life (ca. 3.85 Ga), are very close to the melting temperatures of short-length DNA. An enzyme-free mechanism for replication can therefore be imagined in which the absorption of the energy of UVC photons by the nucleic acids during the day would be sufficient to denature RNA or DNA, allowing the separated strands to act as templates for extension during the cooler periods overnight [4,5]. Such ultraviolet and temperature-assisted replication (UVTAR) bears similarity to the technique of polymerase chain reaction used to amplify particular segments of RNA or DNA in the laboratory [44] but where the temperature cycling is diurnal of reduced amplitude but supplemented with UVC light cycling.

Photon-induced DNA denaturation has, in fact, been experimentally observed [45,46,47]. For short (∼25 bp) synthetic DNA, complete and reversible UVC light-induced denaturing is found even at UVC fluxes a small fraction of what would be expected during the Archean [47]. The rate of denaturing increases with temperature as the melting temperature is approached from below.

The mechanism by which UVC-induced denaturing occurs is unknown. However, based on determinations for synthetic oligos without adjacent thymines, it appears that the rates are significantly higher than those which would be expected due to local denaturing caused by cyclobutane pyrimidine dimer formation or other photoproducts alone [47]. The fact that both pyrimidine dimerization as well as dimer monermization (with, in fact, a larger cross section) occurs within the wavelength range of interest [48] suggests that the formation of dimers, and their subsequent monermization at shorter wavelengths, may have been an operative, although not the only, mechanism of enzymeless UVC-induced denaturing during the Archean. Another process presently being investigated [49] to explain the high observed rates of denaturing is UVC-induced charge transfer [50], which could cause changes in hydrogen bonding between Watson-Crick pairs, thus affecting the charge distributions on the bases and thereby debilitating the stacking interactions between consecutive bases [47].

Diurnal temperature oscillations of the present day sea surface microlayer have been studied by Schlussel [51]. Variations as large as 5 °C have been observed in the North Atlantic [52] due mainly to the absorption of solar infrared light during the day and surface evaporation at night. If such a diurnal oscillation of sea surface temperature also occurred during the Archean, it could have promoted photon-induced enzymeless denaturing of RNA and DNA during the day and extension overnight. Enzymeless extension could have occurred overnight at colder surface temperatures employing Mg^+2^ ions as catalysts [53] and UV-activated phosphorylated nucleotides [54]. Other intercalating planer molecules (for example, the amino acid tryptophan) have been shown to increase the rate of enzymeless extension by orders of magnitude [55].

Enzymes, and thus information content and reproductive fidelity, were not required for replication until the sea surface temperature had cooled to sufficiently below the melting temperature of RNA and DNA. Longer RNA or DNA segments that began to code for (have chemical affinity to) simple denaturing peptides could continue replicating at colder temperatures, thereby initiating information accumulation [56] and evolution through a kind of natural selection based on thermodynamic efficacy of photon dissipation in response to a cooling ocean surface. The driving force for replication, and thus evolution, is the increase in entropy production afforded to the Earth in its interaction with its solar environment [4,5].

## 4. Homochirality through Photon-Induced Denaturing

Scattering of unpolarized UV sunlight from water molecules and suspended particles, and the subsequent total internal reflection of this light at the water–air interface, leads to a component of about 5% circular polarization during late afternoon near the sea surface [9]. The handedness of the circular polarization in the afternoon depends on the hemisphere (for example, right-handed in the Northern hemisphere) due to the particular viewing direction of the sunlight [9] (for example, south to south-west in the afternoon in the Northern hemisphere), of course independently of terrestrial magnetic polarization reversals. Since the sea surface temperature would be greatest in the late afternoon and since higher temperature correlates with higher efficiency for UV-induced denaturing [47], this fact could have contributed to a gradual enhancement of RNA or DNA with D-enantiomer nucleotides because of the greater absorption cross sections of these chiral molecules for right- over left-handed circularly polarized light between about 230 and 270 nm (see Figure 1). Double strands containing L-enantiomer nucleotides would have been at a slight disadvantage for reproduction since they would not quite absorb the right-handed circularly polarized light of the late afternoon as well, and thus could not denature as often. RNA/DNA oligos containing predominantly L-enantiomer nucleotides would thus tend to become locked in the double strand formation, effectively removing them as templates for extension and thus avoiding further reproduction, while D-enantiomer oligos could have continued replicating and thus evolving.

The problem of chiral defects (the polymerized strand containing nucleotides of different chirality) frustrating the extension [2,3] is significantly circumvented in the framework of the present hypothesis since, during the earliest stages of life, environmental conditions rather than enzymes promote replication. In this case, small oligos could potentially replicate without the need to code for enzymes. The chance polymerization of small oligos without a chiral defect is not vanishingly small. For example, the probability of a 10 bp oligo being homochiral is (1/2)^10^ = 9.8 × 10^−4^ (about one in a thousand).

At neutral pH, the circular dichroism (CD) spectrum of DNA shows a negative band (greater absorption of right-handed circularly polarized light) from about 235 nm with a minimum at 245 nm, extending to about 260 nm, and a positive band with a maximum at approximately 275 nm [57]). The negative band has been shown to be a result of base stacking [58] and is relatively independent of base content or sequence (secondary structure), while the positive band depends on these characteristics [57]. The CD spectrum of shorter polynucleotides shows a wider negative CD band spanning the region of 230–270 nm at neutral pH [59] with a minimum at 250 nm (see Figure 1). It is this negative CD band that will be seen to be responsible for the gradual accumulation of homochirality in RNA and DNA through the UVC photon-induced denaturing and temperature-assisted mechanism of replication (UVTAR) described above and analyzed in more detail below.

A quantitative analysis of the rate of chirality acquisition through UVC and surface temperature-induced denaturing requires a convolution of the wavelength-dependent circular dichroism (CD) spectra of the different DNA oligos, their absorption spectrum, and an estimate of the wavelength-dependent intensity of the light reaching Earth’s surface during the Archean. In Figure 2, I plot the circular dichroism for the DNA oligos convoluted with the absorption spectrum of a short (25 bp) oligo [47] and convoluted with an estimate of the light flux arriving at the surface in this wavelength region during the Archean (see Figure 1). From Figure 2, it can be seen that there would have been an important photon absorption differential favoring right-handed oligos in the afternoon (when light was slightly right-handed circularly polarized) or equivalently favoring left-handed oligos in the morning (when light was slightly left-handed circularly polarized). The average value of the circular dichroism over the 200–300 nm region is CD = −0.279, −1.06 and −13.37 for the 20 bp, 11 bp, and adenosine+tryptophan (integrated from 220–30 nm), respectively.

The CD spectrum of RNA and DNA between 200 and 300 nm depends on temperature, salinity, and pH. For small segments of DNA, higher temperature has the effect of reducing the amplitude of both the positive and negative circular dichroism peaks, while increasing the salt concentration increases mainly the negative amplitude with little effect on peak positions or zero crossings [59]. Lowering the pH tends to increase the negative band and to shift it to somewhat longer wavelengths [59]. Ambient pressure would also have a small effect on the CD spectrum. The conditions of the early Archean sea surface are generally considered to have been warmer and more acidic due to higher atmospheric CO_2_ concentration, higher salinity because of a lack of salt capture tidal pools at the shores of newly forming continents, and no more than twice present atmospheric pressures [63]. A truly quantitative analysis of the Archean oligo CD spectrum is not possible until these characteristics become more precisely established; however, since the negative CD band between 230 and 260 nm is practically universal for B-DNA oligos under different environmental conditions [59] and different sizes [57], it is possible to make an approximate determination of the number of naturing/denaturing cycles required for obtaining homochirality through the suggested asymmetric ultraviolet and temperature assisted replication mechanism based on this universal feature of a negative CD band.

## 5. Model Simulations

It is assumed that a racemic population of short oligos of RNA and DNA interacting with amphipathic molecules (having both polar and non-polar ends), produced by UV photochemical dissipative structuring on atmospheric gases [39], floated on the surface of a hot prebiotic Archean ocean. Single-strand RNA or DNA could have begun to act as templates for replication as soon as the sea surface temperature at night dropped below their melting temperature. I now present a simplified model to estimate how rapidly chirality would have grown in a DNA population under these conditions because of an excess of right- over a left-handed circularly polarized light in the late afternoon when surface temperatures were higher and thus more conducive to photon-induced denaturing, considering that right-handed DNA absorbs slightly more of this light than left-handed DNA (see Figure 2).

Since replication could have only begun to occur once the ocean surface temperature at night dropped below the oligo melting temperature (otherwise enzymeless extension could not occur), the oligos that could have replicated would have been on the lower portion of their steep melting curve; therefore, the number of intact hydrogen bonds between the bases would have been a strong function of temperature. Since the probability of complete denaturing of the oligos through the UVC mechanism would have been a function of the number of intact hydrogen bonds, in a first approximation, it is safe to assume that complete denaturing and strand separation would have occurred only in the afternoon when sea surface temperatures were highest and spontaneous renaturation would most likely have been averted. For a given ocean surface temperature, the complete UVC-induced denaturing of DNA, and therefore the possibility for replication, would be, in first approximation, proportional to the amount of UVC light absorbed by the segment. (In reality, a length-dependent energy threshold for denaturing exists due to the hydrogen bonding between complementary strands and stacking energy between consecutive bases. This threshold has important consequences and will be considered in a refinement of the model given below). The following recursion relations then give the number of left-handed and right-handed RNA/DNA oligos ($L_{i}$ and $R_{i}$) at any given diurnal cycle *i*:
(1)$$\begin{array}{c} L_{i}=L_{i-1}\left(1+c\left(P_{LL}+P_{LR}\right)\right)\\ R_{i}=R_{i-1}\left(1+c\left(P_{RR}+P_{RL}\right)\right) \end{array}$$
where *c* is a normalization constant (see below) and $P_{LL}$ and $P_{LR}$ are the average (over all existing strands) relative probabilities that a left-handed DNA absorbs a left- and right-handed photon, respectively. $P_{RR}$ and $P_{RL}$ are similarly defined, but for absorption on a right-handed RNA/DNA.

All probabilities will be taken relative to that of right-handed circularly polarized light absorption on right-handed DNA, $P_{RR}$, this being the largest since most melting would occur in the afternoon when ocean surface temperatures were highest, and afternoon submarine light would have been somewhat right-handed circularly polarized and D-DNA has a net negative circular dichroism band between 230 and 270 nm (see Figure 2) where DNA absorbs strongly. Therefore,
(2)$$\begin{array}{ccc} P_{RR} & = & 1.0\\ P_{RL} & = & P_{RR}\cdot \frac{\left(1.0-\Delta_{RLE}\right)}{\left(1.0+\Delta_{RLE}\right)}\cdot \frac{\left(1.0-\Delta_{RCD}\right)}{\left(1.0+\Delta_{RCD}\right)}\\ P_{LL} & = & P_{RR}\cdot \frac{\left(1.0-\Delta_{RLE}\right)}{\left(1.0+\Delta_{RLE}\right)}\\ P_{LR} & = & P_{RR}\cdot \frac{\left(1.0-\Delta_{RCD}\right)}{\left(1.0+\Delta_{RCD}\right)} \end{array}$$
where $\Delta_{RLE}$ is the right-handed circular polarized light excess of the afternoon, and $\Delta_{RCD}=-\left(\epsilon_{L}-\epsilon_{R}\right)/\left(\epsilon_{L}+\epsilon_{R}\right)=-g/2$, with *g* as the absorption anisotropy factor (see Section 2). The $\Delta_{RLE}$ of submarine light at the sea surface may be as high as 10% [9], but I take a value of 5% as a more conservative estimate. According to Gray et al. [59], the anisotropy factor g/2 due to the differential molar absorption of right- over left-handed photons at 250 nm for short polynucleotides is about 4/6000. However, this is a function of wavelength, and it is possible to obtain a better value of the wavelength average by integrating the convoluted circular dichroism, Figure 2, over wavelength. Integrating from 200 to 300 nm gives an average CD of −0.279 and −1.06 [M^−1^ cm^−1^] for the 20 bp oligo and the 11 bp oligo, respectively. The integration of the total absorption of the oligos (dotted black curve in Figure 1) over the 200–300 nm range gives $\epsilon_{T}=\epsilon_{L}+\epsilon_{R}=4469$ [M^−1^ cm^−1^]. Therefore,
(3)$$\Delta_{RCD_{11bp}}=-\frac{\left(\epsilon_{L}-\epsilon_{R}\right)}{\left(\epsilon_{L}+\epsilon_{R}\right)}=-\frac{CD}{\epsilon_{T}}=\frac{1.06}{4469},$$
and, similarly,
(4)$$\Delta_{RCD_{20bp}}=\frac{0.279}{4469}.$$


The normalization constant *c* in Equation (Equation 1) would depend on both daytime denaturation factors (intensity of the UV light, sea-surface temperature, pH, salinity, etc.) and nighttime extension factors (the length of DNA strand, the concentration of activated nucleotides available at the sea-surface, the sea-surface temperature at night, duration of night, etc.). In lieu of better constrained data for the Archean, and for the sake of calculation, I consider short oligos of ∼10 nucleotides in length and take, as a plausible value for *c*, 0.001, i.e., only one in 1000 DNA oligos reproducing through the UVTAR mechanism during each diurnal cycle. (For example, this compares to the probability that 10 nucleotides chosen at random from a racemic solution would have the same chirality, i.e., ${\left(0.5\right)}^{10}\approx 0.001$. As another example, in a polymerase chain reaction, with an unlimited supply of nucleotides, primers, and the enzyme polymerase, and extension times of the order of minutes, the value of *c* is very close to 1).

Using these values in the recursion Equation (Equation 1) together with the following equation for the chirality $C_{i}$ as a function of diurnal cycle *i*,
(5)$$C_{i}=-\frac{L_{i}-R_{i}}{L_{i}+R_{i}}$$
gives the three curves (a) plotted in Figure 3. The curves are independent of the initial population values $L_{0},R_{0}$ as long as the population is initially racemic, i.e., $L_{0}=R_{0}$.

Since an Archean day was about 1/2 the length of an actual day, and assuming the orbit of Earth has not changed since the Archean, Figure 3a implies that 100% homochirality can be obtained in approximately 5 million Archean years for the 20 bp oligo with a small circular dichroism (−0.279 M^−1^ cm^−1^) and 164 thousand years for the 11 bp oligo with a larger circular dichroism (−1.06 M^−1^ cm^−1^).

The model can be made more realistic by including stochastic fluctuations of the probabilities for photon absorption on the different chirality oligos to mimic the statistical fluctuations involved in photon absorption and including an energy absorption threshold for denaturation. Such an energy threshold exists due to the specific temperature-dependent hydrogen bonding and stacking energies between the two complementary strands. The threshold would be larger in the morning than in the afternoon because of a cooler surface temperature. Denaturation of right-handed RNA or DNA would therefore be further favored by this threshold since in one hemisphere (Northern) there was an excess of right-handed circularly polarized light available in the afternoon when surface temperatures were warmest. This stochastic fluctuation in absorption probability and energy threshold were included in the model by randomly varying the $P_{RR},P_{LL}$, etc. by (±2%) about their nominal values given by Equation (Equation 2), and setting the combined probabilities $(P_{LL}+P_{LR})/2$ or $(P_{RR}+P_{RL})/2$ to 0.6 if they fall below a specified limit (0.98). Such a threshold dramatically increases the rate of obtainment of homochirality, giving times as short as 3000 Archean years for the short 11 bp oligo and 13,000 years for the 20 bp oligo (blue and black curves (b) of Figure 3, respectively).

Increasing the replication rate from 0.1% to 1.0% per diurnal cycle, *c* = 0.01, leads to the obtainment of homochirality for the 11 bp oligo in only 230 Archean years (not shown), considering the energy threshold defined above. A greater right-handed circular polarization of submarine UVC light during the afternoon, instead of that assumed of 5%, would further reduce this time.

Finally, it is interesting to consider the role that tryptophan may have played in fomenting homochirality through enzymeless replication early in the history of life. Tryptophan absorbs strongly in the long-wavelength UVC region that was arriving at Earth’s surface during the Archean, but since it does not possess a conical intersection, it decays on nanosecond time scales and fluoresces [64]. However, when coupled to RNA or DNA, it can pass its excitation energy to the nucleic acid through resonant energy transfer, including photon-induced proton transfer, leading to complete quenching of its fluorescence [65]. Tryptophan could thus have acted as an antenna molecule to foment photon dissipation, and this characteristic alone would have provided sufficient thermodynamic justification for its association with nucleic acid. However, energy and charge transfer could also have augmented the probability of photon-induced denaturing of RNA or DNA in cooler ocean waters. Furthermore, tryptophan is an amphiphilic molecule that would have prevented sedimentation of DNA and RNA, thereby exposing these molecules to the most intense flux of UVC light. Finally, enzymeless extension is increased by orders of magnitude when tryptophan is present [55].

Perhaps the most important advantage of tryptophan to the UVTAR mechanism, however, is related to the fact that the negative circular dichroism of D-nucleosides in the spectral region of interest is greatly enhanced in the presence of L-tryptophan (see Figure 1 and Figure 2). The same is true when L-tryptophan is present with DNA [62], the negative CD band of DNA appearing to be simply an average of all the involved nucleoside bands. Using the circular dichroism of adenosine with tryptophan, convoluted over the spectral region of relevance, giving a value of −13.37 M^−1^ cm^−1^ (see Figure 2) as a proxy for DNA with tryptophan, gives, using Equation (Equation 2) with Equation (Equation 1), an approximate time to homochirality of 13,000 Archean years, and only 273 years if the existence of an energy threshold to denaturing is considered (see previous section and red curves of Figure 3). Although there is some indirect evidence that tryptophan may not have been among the original amino acids of life [8], of all the amino acids, tryptophan has the strongest affinity to its anticodon, and this may be evidence for its participation in a very early stereochemical era [66].

Until data become available on the circular dichroism of other amino acids with DNA, this proposal for the production of homochirality cannot be generalized to all amino acids. However, there does exist some data suggesting that other amino acids besides tryptophan may have acquired homochirality through a similar phenomena. As ocean surface temperatures cooled and the ocean’s salinity increased, longer RNA or DNA segments would spontaneously take on cholesteric mesophases [67] in which the right-handed double-helix folds in on itself to produce a supra-molecule with enhanced right-handed chirality. The circular dichroism of these long cholesteric forms is, however, positive within the 200–300 nm region [67] and these by themselves would therefore not absorb as well the right-handed circularly polarized light of the afternoon. However, Reich et al. [67] have also shown that some L-peptides (poly Lys and poly Lys-Ala) have a significantly larger affinity to D-DNA in the cholesteric form than do D-peptides and that these D-DNA+L-peptide complexes have, in fact, a negative circular dichroism over the whole 200–300 nm region [67]. These D-DNA+L-peptide complexes would then be more efficient at absorbing right-handed circularly polarized photons of the afternoon and therefore have greater probability of denaturing through the UVC photon-induced mechanism, thus providing new templates for overnight extension.

## 6. Conclusions

I have proposed a hypothesis for the obtainment of homochirality through an enzymeless RNA or DNA replication mechanism, promoted by an asymmetric right- over left-handed UVC photon-induced denaturing of RNA or DNA due to a negative circular dichroism band extending from 230 to 270 nm for small oligos and an excess of right-handed circular polarized submarine light at the sea surface in the afternoon when surface temperatures were highest and thus most conducive to denaturing. This proposal for homochirality has advantages over either photoreaction, photocatalyzing, or photolysing because, as in the polymerase chain reaction, the effect operates exponentially on chirality (or any other characteristic relevant to replication) with each diurnal cycle. No destruction of the original material is necessary; indeed, through natural racemization, nucleotides of initially inappropriate chirality will eventually become usable. It also deals with weak hydrogen bonds between strands and intra-base stacking interactions rather than strong covalent bonds that must be ruptured in photolysing or photoreactions. Furthermore, since this mechanism would have operated below the melting temperature of RNA or DNA, there exists a temperature-dependent energy absorption threshold related to the strength of these bonds, which becomes greater as the sea surface cools, thus more strongly favoring D-RNA or D-DNA.

A plausible scenario for the homochirality of some amino acids is that of chiral selectivity of D-nucleic acid for L-amino acids due to complimentary stereochemistry, particularly when DNA is in its folded cholesteric form, of relevance to longer-strand RNA or DNA. This, in turn, has relevance to lower sea surface temperatures after the origin of life, when small peptides or enzymes to aid replication became necessary [4,5,56]. Some D-DNA+L-peptide complexes have a negative circular dichroism over the entire 200 to 300 nm region, whereas D-DNA+R-peptide complexes have a positive circular dichroism over this region [67]. D-DNA+L-peptide complexes would thus have had somewhat greater replication probability under the UV and temperature-assisted replication scenario.

The mechanism postulated here for the procurement of homochirality should be considered as a far-from-equilibrium process operating under cyclical boundary condition, rather than a non-equilibrium spontaneous symmetry breaking process. The mechanism is dependent on photon absorption and dissipation into heat. The dissipation of a generalized thermodynamic potential is, in fact, what drives all irreversible processes.

An experimental test of this hypothesis would be to measure the differential polymerase chain reaction rates of D-DNA at temperatures somewhat below their melting temperatures with only weak temperature cycling under right- and left-circularly polarized UVC light. The effect of including tryptophan on this differential rate should also be considered.

## Figures and Tables

**Figure 1 life-08-00021-f001:**
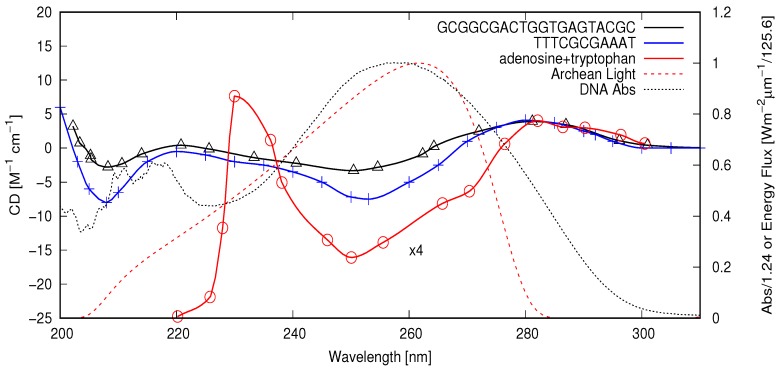
Circular dichroism of the 20 base pair oligo B-form duplex 3′-GCGGCGACTGGTGAGTACGC (M.W. 12,306 g mol^−1^) at neutral pH and room temperature taken from Figure 3a of [60] (black line with triangles). Circular dichroism of the 11 base pair hairpin B-form duplex 3′-TTTCGCGAAAT (M.W. 6839 g mol^−1^) (blue line with crosses, left y-scale) at neutral pH and room temperature adapted from Figure 5a of [61]. Circular dichroism of adenosine with tryptophan in the ratio 1:2 taken from Figure 3a of Arcaya et al. [62] (red line with circles: the CD spectrum for adenosine+tryptophan is plotted divided by 4 for convenience of scale. Arcaya et al. state that the CD spectrum of DNA+tryptophan is similar to the average of the CD spectra of the individual nucleosides with tryptophan [62]). Short oligo (25 bp) absorption spectrum taken from [47] normalized to a peak value of one (black dotted line, right y-scale). Note that, at wavelengths shorter than 220 nm, the absorption spectrum is not reliable due to the reduced output of the deuterium light used [47]. Solar Archean spectrum at Earth’s surface [40] normalized to a peak value of one (red dashed line, right y-scale).

**Figure 2 life-08-00021-f002:**
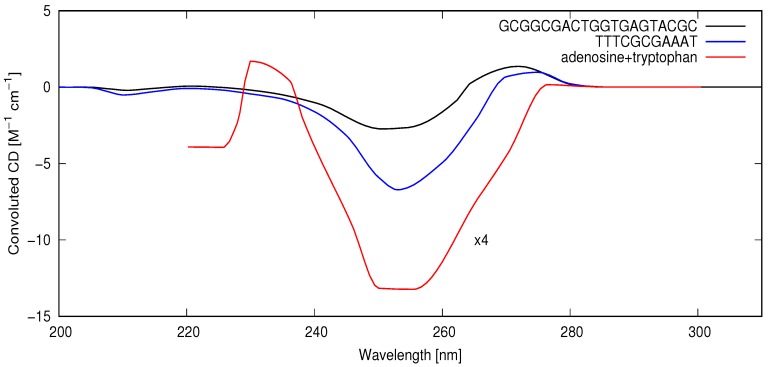
Circular dichroism spectra of the 20 bp oligo 3′-GCGGCGACTGGTGAGTACGC, the 11 bp oligo 3′-TTTCGCGAAAT, and adenosine+tryptophan convoluted with a short oligo (25 bp) absorption spectrum and the wavelength-dependent intensity of the light reaching Earth’s surface during the Archean (see Figure 1). Adenosine+tryptophan is plotted at 1/4 of its real amplitude for convenience of scale. The average CD values, integrated over the region 200 to 300 nm are −0.279, −1.06 and −13.37 (220 to 300 nm) for 3′-GCGGCGACTGGTGAGTACGC, 3′-TTTCGCGAAAT and adenosine+tryptophan respectively.

**Figure 3 life-08-00021-f003:**
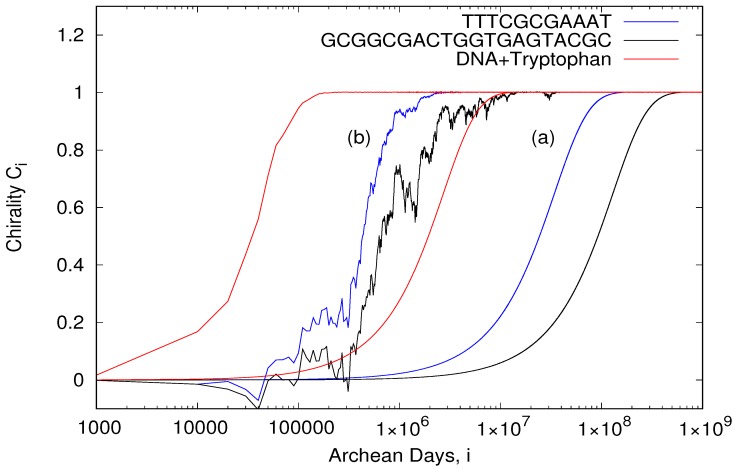
Chirality *C_i_* as a function of the number of Archean days *i* for the two oligos and DNA+Tryptophan as calculated through Equation (Equation 5). (a) The three smooth curves to the right were calculated assuming denaturation probability is simply proportional to the number of photons absorbed, but that complete denaturing occurs only in the afternoon, and a 5% excess of right-hand circularly polarized submarine light at the ocean surface during the afternoon. (b) These three curves to the left include in the calculation a stochastic photon absorption probability (±2% with respect to nominal values given by Equation (Equation 2)) and an energy threshold for denaturation related to the complementary strand binding energy (see text).

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
