# Peer review of "Homochirality through Photon-Induced Denaturing of RNA/DNA at the Origin of Life"

_life, 2018, doi:10.3390/life8020021_

Round 1

Reviewer 1 Report

This manuscript provides some (interesting) discussion on the sources of enantiomeric excess in biomolecules, oligonucleotides like RNA/DNA, caused by right- or left-circularly polarized photons in prebiotic scenarios such as submarine niches subjected to thermal fluctuations, and hence triggering selective photon-induced denaturation of the above-mentioned biomolecules. Rather than a novel theory or mechanism accounting for the origin of homochirality, it represents a working hypothesis that will require further experimental evidences. In principle, that hypothesis might have worked and this referee finds no flaws in the core argument. Still, like in other photochemically-induced processes, the protocol would require amplification cycles to achieve a significant enantiodiscrimination. That said, this contribution also raises a series of thermodynamic and kinetic difficulties (see below).

The manuscript is organized in different subsections. Section 2 (on previous homochirality theories) is a good, well-referenced overview, albeit largely confined to photochemical processes. The author ignores other more plausible models of symmetry breaking such as the spontaneous crystallization and resolution of conglomerates, as well as polymerization models, which are not discussed at all. However, in the context of this manuscript (another photochemical hypothesis), that overview probably suffices.

The author invokes the thermodynamic dissipation theory, which is unclear to me, as only two self-citations (Refs. 3 and 4) are provided (one being a preprint in the arXiv server, dated 2009). I suppose the author refers probably to the classical thermodynamic arguments in dissipative systems formulated by Prigogine (and Prigogine-Kondepudi) in the 1970s. Another self-edited book (Ref. 36) apparently discusses that theory. Moreover, the other self-citation (Ref. 45) related to the present work is also a preprint (bioRxiv) and therefore, it is difficult to assess the validity of the present hypothesis within the scientific community.

I would give this paper a chance, if the author were able to re-formulate the photochemical story as a working hypothesis comprising the interaction of CPL in biomolecules, at least from data in Figures 1 and 2. As mentioned, there are other issues the author overlooks to a significant extent. Thus, how were RNA or DNA strands generated from a random mixture of nucleotides? The author does assume the prebiotic formation of such biomolecules and ignores the problems of complexity in polymerization reactions (an excellent and highly-cited paper by Avetisov and Goldanskii, PNAS 1996, 93, 11435-11442 deals with symmetry breaking and formation of homochiral sequences).

Regarding the simulation model, the author likewise assumes that complete melting and strand separation would have ocurred only in the afternoon at higher surface temperatures. I do not understand why should that assumption be true.

Finally, the case of tryptophan is misleading and, in my opinion has little to do with the first part of this manuscript. Although I agree with conclusions arising from interactions between biomolecules (like oligonucleotides and poly-lysine as mentioned by Michaelian), such processes are obviously post-evolutionary steps. On the other hand, there was some controversial discussion in the past concerning the action of photo- and radiolytic processes on tryptophan enantiomers (e.g. Nature 1976, 261, 522-524; Nature 1979, 281, 151; J. Mol. Evol. 1980, 15, 21-28). Further studies clearly suggest that complex amino acids such as tryptophan were recruited late in proteins (for a good monograph that reports detailed information on prebiotic and interstellar amino acids: U. Meierhenrich, Amino Acids and the Asymmetry of Life, Springer, 2008, Sect. 7). 

Finally, if one considers a photochemical transformation with biomolecules, side reactions are unavoidable. Generation and propagation of radical species, which would have caused structural damage, tautomerization and isomerization within the RNA/DNA strands complicate the scenario still further.

The author should seriously re-assess the present hypothesis, paying attention to additional literature. Some typos should be corrected through the entire contribution.

Author Response

I thank the referee for their thorough and useful review of my manuscript. In the following I list the referee’s comments followed by my responses to each.

Referee: “This manuscript provides some (interesting) discussion on the sources of enantiomeric excess in biomolecules, oligonucleotides like RNA/DNA, caused by right- or left-circularly polarized photons in prebiotic scenarios such as submarine niches subjected to thermal fluctuations, and hence triggering selective photon-induced denaturation of the above-mentioned biomolecules.”

“Rather than a novel theory or mechanism accounting for the origin of homochirality, it represents a working hypothesis that will require further experimental evidences. In principle, that hypothesis might have worked and this referee finds no flaws in the core argument. Still, like in other photochemically-induced processes, the protocol would require amplification cycles to achieve a significant enantiodiscrimination. That said, this contribution also raises a series of thermodynamic and kinetic difficulties (see below).”

Response: In the revised version of the manuscript I present the proposal as a hypothesis and suggest how it could potentially be experimentally falsified. I further emphasize that it relies on the observed phenomena of photon-induced denaturing for which I cite experimental evidence from three independent investigations. Reference to the thermodynamic theory of the origin of life through dissipative structuring is not required and I now only bring it into the discussion when denaturing is put into the broader context of enzyme-less replication.

I thank the referee for their encouragement regarding their evaluation of the core argument.

To function, the hypothesis does not require any initial entatiomer enrichment, and therefore does not require any amplification of such an initial enrichment. However, there is differential amplification operating due to the diurnal (morning/afternoon) temperature and UVC light circular polarization asymmetry, leading to preferential denaturing of oligos, or segments of oligos which are homochiral and of a specific chirality (see below). Those of mixed or opposite chirality would either not be extended or would be less likely to denature (lower differential reproduction). The hypothesis works with an initial racemic distribution of nucleotides.

Referee: “The manuscript is organized in different subsections. Section 2 (on previous homochirality theories) is a good, well-referenced overview, albeit largely confined to photochemical processes. The author ignores other more plausible models of symmetry breaking such as the spontaneous crystallization and resolution of conglomerates, as well as polymerization models, which are not discussed at all. However, in the context of this manuscript (another photochemical hypothesis), that overview probably suffices.”

Response: I thank the referee for their positive analysis of Section 2. I will not include (unless the referee considers it necessary) a discussion of spontaneous crystallization and selective adsorption in this overview section since, as the referee suggests, these are somewhat removed from the hypothesis and they are in fact mentioned in the review by Podlech (2001) which I reference at the beginning of section 2 for those readers wanting a more comprehensive review of the homochirality problem. I now also include a reference to the interesting book by Meierhenrich (2008) in this regard. Polymerization is now discussed but only with regard to chiral defects (see below).

Referee: “The author invokes the thermodynamic dissipation theory, which is unclear to me, as only two self-citations (Refs. 3 and 4) are provided (one being a preprint in the arXiv server, dated 2009). I suppose the author refers probably to the classical thermodynamic arguments in dissipative systems formulated by Prigogine (and Prigogine-Kondepudi) in the 1970s. Another self-edited book (Ref. 36) apparently discusses that theory. Moreover, the other self-citation (Ref. 45) related to the present work is also a preprint (bioRxiv) and therefore, it is difficult to assess the validity of the present hypothesis within the scientific community.”

Response: Although we have published other papers, not cited in the present manuscript, which suggests an active consideration of the theory by the scientific community, in the new version of the manuscript, I now only make reference to the experimentally observed photon-induced denaturing of DNA and give reference to three independent experiments. It is therefore no longer necessary for the reader to assess the validity of the dissipation theory, but rather only the validity of photon-induced denaturing. The foundations of the dissipation theory for the origin of life are indeed derived from Classical Irreversible Thermodynamic Theory in the non-linear regime as developed by Onsager and Prigogine and others.

Referee: “I would give this paper a chance, if the author were able to re-formulate the photochemical story as a working hypothesis comprising the interaction of CPL in biomolecules, at least from data in Figures 1 and 2.”

Response: I have re-formulated the theory as a working hypothesis through implementation of the changes described above and below. I am grateful to the referee for the chance for the proposal to be evaluated by the community at large.

Referee: “As mentioned, there are other issues the author overlooks to a significant extent. Thus, how were RNA or DNA strands generated from a random mixture of nucleotides? The author does assume the prebiotic formation of such biomolecules and ignores the problems of complexity in polymerization reactions (an excellent and highly-cited paper by Avetisov and Goldanskii, PNAS 1996, 93, 11435-11442 deals with symmetry breaking and formation of homochiral sequences).”

Response: I thank the referee for bringing my attention to this very interesting paper which I now cite in the new version of the manuscript in the context of the problem of polymerization. The referees question is, at least to some degree, relevant for all hypotheses regarding the origin of homochirality. However, an important point can be made in favor of the proposed hypothesis with respect to this difficult problem. For replication to be operative at low temperature and under some other generalized thermodynamic potential (not the UVC potential), it would have to be assumed that the oligos had what Avetisov and Goldanskii refer to as “biological significance”, that is, that they were long enough to code for simple enzymes that could aid with denaturing and extension. They suggest this size to be N > 150, which, as they point out, would have had a vanishingly small probability of being polymerized without a chiral defect. However, in the framework employed for the presented hypothesis, environmental conditions are assumed to replace the action of the replication enzymes; in particular, UVC light and temperature for denaturing, and UVC phosphorylation and UVC polymerization plus Mg+2 catalysis for extension. In this case, small oligos could potentially replicate without enzymes. The chance polymerization of small oligos without a chiral defect is not vanishingly small. In the new version of the manuscript, I give the example of a 10 bp oligo for which the probability of obtaining a homochiral polymerization is (1/2)^10=9.8^-4 (about one in a thousand). This number I use as the constant “c” in equation (1) of my manuscript and this gives rise to the chirality curves that are plotted in figure 3. Smaller oligos would reach homochirality faster and these could have been used for the extension of longer oligos. A new paragraph describing this is given in section 4 along with reference to the Avetisov and Goldanskii paper.

Referee: “Regarding the simulation model, the author likewise assumes that complete melting and strand separation would have ocurred only in the afternoon at higher surface temperatures. I do not understand why should that assumption be true.”

Response: The hypothesis assumes that replication could only begin to occur once the ocean surface temperature at night dropped somewhat below the denaturing temperature, otherwise enzyme-less extension could not occur. At these temperatures, the oligos would be at the lower portion of their steep melting curve. To give an example, assume that at the beginning of life during the morning until midday, the sun heated the ocean surface to the melting temperature of the oligo. At this temperature, 50% of the base pairs on average would be denatured. The probability that a given 10 bp oligo would be completely denatured would be roughly (0.5)^10 = 9.8x10^-4 (about 1 in a thousand). For the sake of illustration, suppose that for UVC light-induced denaturing to work (i.e. to completely separate the strands with UV light) during the morning (or afternoon) it requires a proportion, say at least 7 out of the 10 base pairs, to be already denatured by the temperature of the ocean surface. The probability of finding a given oligo not completely denatured but with at least 7 Watson-Crick base pairs denatured would be approximately (0.5)^7=.0078 (about one in 128). Depending on the steepness of the temperature melting curve, this ratio would increase rapidly as the surface temperature continued to warn up during the afternoon (say 5°C warmer at 15:00 hr, giving say 70% denaturing) the probability of finding a completely separated oligo would then be (0.7)^10 = 0.028 (one in 35). The probability of finding an oligo not completely separated into single strands but with at least 7 base pairs denatured would be approximately (0.7)^7=0.082 (one in 12). Therefore, the comparison is between one in 128 oligos that would denature through this UVC mechanism during the morning compared with one in 12 that would denature during the afternoon (an order of magnitude greater). For this reason it is safe to neglect denaturing during the morning. I have now included a few new sentences in section 5 of the manuscript justifying this assumption.

Referee: “Finally, the case of tryptophan is misleading and, in my opinion has little to do with the first part of this manuscript. Although I agree with conclusions arising from interactions between biomolecules (like oligonucleotides and poly-lysine as mentioned by Michaelian), such processes are obviously post-evolutionary steps.”

Response: I believe that an explanation, from within the same framework, for the homochirality of at least some of the amino acids would lend much support to the proposed mechanism for homochirality. For reasons given below, at least for the case of tryptophan, I believe that it is not obviously the case that the complexification of RNA or DNA with complex amino acids necessarily has to be considered as post-evolutionary.

Referee: “On the other hand, there was some controversial discussion in the past concerning the action of photo- and radiolytic processes on tryptophan enantiomers (e.g. Nature 1976, 261, 522-524; Nature 1979, 281, 151; J. Mol. Evol. 1980, 15, 21-28).”

Response: These articles all refer to induced optical activity in the amino acids due to beta or light radiation (ostensibly through beta- or photo- lysing). As such, they do not seem to have much relevance to my present proposal which considers the induction of optical activity in DNA+amino acid complex due to stereochemical effects resulting from the interaction between the two molecules. Differential proliferation of these due to differential UVC-induced denaturing of the complex is assumed as for the case of DNA or RNA alone.

Referee: “Further studies clearly suggest that complex amino acids such as tryptophan were recruited late in proteins (for a good monograph that reports detailed information on prebiotic and interstellar amino acids: U. Meierhenrich, Amino Acids and the Asymmetry of Life, Springer, 2008, Sect. 7).”

Response: I thank the referee for this interesting and complete reference which is now included in the new version of the manuscript in the context of a valuable review of homochirality (particularly with respect to amino acids) and as a reference for the origin of the complex amino acids. However, not everyone is in agreement with the suggestion that the complex amino acids such as tryptophan were recruited late. For example, of all the amino acids, tryptophan has the strongest chemical affinity to its anitcodon and was therefore suggested to be part of a very early stereochimical era for the genetic code (Yarus et al. 2009). A new sentence has been included at the top of page 12 to emphasize this. Notwithstanding the lack of identification of a route to the production of tryptophan on Earth’s surface or atmosphere from primordial molecules, tryptophan appears to have a similar structural complexity as that of the nucleotides. Although the actual biosynthetic pathways to produce complex molecules using the visible light of today are complex, it may not have been so during the Archean when UVC light was available, an example of this is the UVC production from HCN of the nucleobase adenine (Michaelian, 2017). That these complex amino acids are not detected in interstellar space or in meteorites but were, perhaps, formed on early Earth, may have to do with the region of the UV spectrum available for construction or lysing. For example, wavelengths shorter than 210 nm can dissociate many carbon based organics and while this light would have been present in a space environment, it was not present on the surface of early Earth (Sagan, 1973).

Referee: “Finally, if one considers a photochemical transformation with biomolecules, side reactions are unavoidable. Generation and propagation of radical species, which would have caused structural damage, tautomerization and isomerization within the RNA/DNA strands complicate the scenario still further.”

Response: Organic molecular dissociation cross sections are generally small within the 210 to 290 nm UVC region relevant to the present hypothesis. Without the presence of molecular oxygen, the formation of radical species would be limited. Both RNA and DNA are very robust to this region of UVC light. The nucleobases have conical intersections that provide sub picosecond de-excitation, allowing essentially no time for chemical reactions (Pecourt et al., 2000). Important lesions, however, occur on stacked bases which may form cyclobutane pyrimidine dimers (CPDs). These are formed with maximum cross section at 255 nm, however, these are not permanent lesions since they can be monomerized with light of 238 nm (Garcès and Dávila, 1981). The monomerization cross section is, in fact, larger than the dimerization cross section leading to a low concentration of dimer lesions in the stationary state. Another decay channel of photoexcited DNA leads to reactive charge transfer states and these may be responsible for the monomerization (Bucher et al., 2016).

Tryptophan, although not having a conical intersection itself, can intercalate between the bases and pass its excitation energy through resonant energy transfer to the nucleobase after which the excitation energy decays through the conical intersection of the nucleobase, thereby greatly reducing the potential for side chemical reactions. Intercalation of tryptophan also reduces significantly the UVC formation of dimers on DNA (Arcaya et al., 1971). As the experiments by Arcaya et al. (1971) listed in the manuscript show, there appears to be no significant structural damage to the tryptophan-DNA complex under the relevant wavelength region of UVC light (210-290 nm).

Referee: “The author should seriously re-assess the present hypothesis, paying attention to additional literature. Some typos should be corrected through the entire contribution.”

Response: I thank the referee for their comments. I have revised the manuscript in accordance with these comments and my responses as listed above. I have included reference to new literature cited by the referee. I believe that all typos have been corrected. Numerous small corrections to improve the redaction were made without affecting content.

References:

Arcaya G, Pantoja ME, Pieber M, Romero C, Tohá JC. (1971) Molecular Interaction of L-Tryptophan with Bases, Ribonucleosides and DNA.  Z Naturforsch B 10, 1026-1030.

Bucher, D.B., Kufner, C.L., Schlueter,A. Carell, T. and Zinth, W.  (2016) UV-Induced Charge Transfer States in DNA Promote Sequence Selective Self-Repair, J. Am. Chem. Soc.  138, 186−190. DOI: 10.1021/jacs.5b09753

Garcès F. & Davila C. A. (1982) Alterations in DNA irradiated with ultraviolet radiation---I. The formation process of cyclobutylpyrimidine dimers: cross sections, action spectra and quantum yields. Photochem. Photobiol. 35, 9-16.

Pecourt. J.M., Peon, J. & Kohler, B. (2000) Ultrafast internal conversion of electronically excited RNA and DNA nucleosides in water. J. Am. Chem. Soc. 122, 9348-9349.

Sagan, C. (1973) Ultraviolet Selection Pressure on the Earliest Organisms, J. Theor. Biol., 39, 195--200.

Yarus, M., Widmann, J.J., Knight, R. (2009) RNA-amino acid binding: a stereochemical era for the genetic code, J. Mol. Evol., 69(5), 406-429.

Reviewer 2 Report

With regards to the expermental methods, the manuscript suffers considerably.

The detailds are almost completely lacking.

I recommend the manuscript can be published in Life.

Author Response

I thank the referee for their review of my manuscript and for their recommendation to publish. The referee’s comments and my responses to them appear below.

Referee: “With regards to the expermental methods, the manuscript suffers considerably. The detailds are almost completely lacking.”

Response: In the new version of the manuscript, emphasis is now placed on UVC- induced denaturing rather than the thermodynamic dissipation theory. Reference is made to three articles giving experimental evidence for UVC- induced denaturing of DNA. Two of these articles are published, the other is an experiment carried out by the present author and a collaborator which is under review. The details of the experimental methods used in this latter paper can be obtained from the bioRxiv reference given in the manuscript. At the end of the Conclusions section, I suggest a PCR type experiment to test the overall hypothesis.

Referee: “I recommend the manuscript can be published in Life.”

Response: I thank the referee for their recommendation to publish.

Round 2

Reviewer 1 Report

The revised manuscript has been, in general, clarified and improved in some instances. The author has replied in detail every point supplying, if pertinent, additional information and citations, which help re-assessment. (*Before going any further, PLEASE note the following: the bibliographic section was missing and citations are shown in the main text as quotation marks-the revised file has 13 pages versus 15 pp of the former version. Not sure it the downloaded manuscript comes from a poor file-to-file conversion).

After a second assessment, I have a few remarks without denying the potential value of this hypothesis. In my opinion evolutionary constraints are compulsory; otherwise claims about Archaen periods or prebiotic scenarios make no sense. Clearly, the present hypothesis has little to do with the origin of life, but rather with the possibility of generating enantioenrichment with/on pre-formed oligonucleotides, regardless of their chemical origin. Still, my main criticism is related to the homochirality of amino acids. It is enormously speculative to justify a potential source of homochirality based on this single example with tryptophan. Even if one assumes a pre-evolutionary formation of tryptophan (in terms of the new references provided in author’s reply), I wonder how could the conjecture be extrapolated to other amino acids in the absence of evidences. In short, this elicits an unnecessarily controversial look at the paper. In fact, section 6 might be removed as independent section because it does not affect the photochemical hypothesis. The DNA/RNA-tryptophan interaction could then be briefly mentioned as in vitro experiment within the context of cholesteric (chiral) phases, which are of course important in asymmetric amplification. Abstract and Conclusions are too long and should keep this at a mínimum. 

On the other hand, and again within the evolutionary framework (consistent with the RNA world), one should assume that RNA (and DNA) would have undergone compartmentalization prior to further chemical transformations and information storage. It would be more realistic the existence, in a submarine environment, of random oligonucleotides whose chirality could have been selected (perhaps via crystallization) before the appearance of RNA sequences with enzyme-like activity.

A very minor extra remark: for the sake of clarity, it would be interesting to mention the range of MW (molecular mass or average molecular mass) of oligos comprising a certain bp (base pairs) size.

Author Response

I once again thank the referee for their time and useful comments in reviewing my manuscript. I have revised the manuscript according to their suggestions. The latest changes to the manuscript are emphasized in blue. Below are the referee’s comments and my responses to each of them.

Referee: “The revised manuscript has been, in general, clarified and improved in some instances. The author has replied in detail every point supplying, if pertinent, additional information and citations, which help re-assessment. (*Before going any further, PLEASE note the following: the bibliographic section was missing and citations are shown in the main text as quotation marks-the revised file has 13 pages versus 15 pp of the former version. Not sure it the downloaded manuscript comes from a poor file-to-file conversion).”

Response: It seems that, for some reason, the LaTex compilation of my supplied files by the copy editor was not completely successful. It may have been due to a conflict in naming the new .bib file. In my letter to the editor I have asked him to ensure that the copy editor carefully revise the .pdf file before sending it to the referee.

Referee: “After a second assessment, I have a few remarks without denying the potential value of this hypothesis. In my opinion evolutionary constraints are compulsory; otherwise claims about Archaen periods or prebiotic scenarios make no sense. Clearly, the present hypothesis has little to do with the origin of life, but rather with the possibility of generating enantioenrichment with/on pre-formed oligonucleotides, regardless of their chemical origin. Still, my main criticism is related to the homochirality of amino acids. It is enormously speculative to justify a potential source of homochirality based on this single example with tryptophan. Even if one assumes a pre-evolutionary formation of tryptophan (in terms of the new references provided in author’s reply), I wonder how could the conjecture be extrapolated to other amino acids in the absence of evidences. In short, this elicits an unnecessarily controversial look at the paper. In fact, section 6 might be removed as independent section because it does not affect the photochemical hypothesis. The DNA/RNA-tryptophan interaction could then be briefly mentioned as in vitro experiment within the context of cholesteric (chiral) phases, which are of course important in asymmetric amplification. Abstract and Conclusions are too long and should keep this at a mínimum.”

Response: I am in agreement with the referee. I have now removed section 6 and instead only mention the possibility of generating the homochirality of tryptophan through the proposed mechanism involving the stereochemical affinity of tryptophan to DNA and RNA and the CD spectrum of the complex. I further mention that until the relevant CD data become available, this particular example cannot be generalized to the other amino acids. The abstract has also been revised to reflect this. However, I do suggest that some generalization may still be possible in the context of the cholesteric mesophases of RNA and DNA for which there does exist some experimental evidence for L-peptides of other amino acids (poly Lys and poly Lys-Ala) with stereochemical affinity and fomenting a negative circular dichroism for the complex over the 200 to 300 nm region (Reich et al., 1996).

The Abstract and Conclusions have been shortened.

Referee: “On the other hand, and again within the evolutionary framework (consistent with the RNA world), one should assume that RNA (and DNA) would have undergone compartmentalization prior to further chemical transformations and information storage. It would be more realistic the existence, in a submarine environment, of random oligonucleotides whose chirality could have been selected (perhaps via crystallization) before the appearance of RNA sequences with enzyme-like activity.”

Response: This is consistent with the proposal presented in the manuscript. Other mechanisms for the selection of chirality may have been operating.

Referee: “A very minor extra remark: for the sake of clarity, it would be interesting to mention the range of MW (molecular mass or average molecular mass) of oligos comprising a certain bp (base pairs) size.”

Response: I have now given the average molecular weights of the oligo sizes at their first appearance in the manuscript, in figure 1.

Bibliography:

Reich, Z.; Schramm, O.; Brumfeld, V.; Minsky, A. Chiral discrimination in DNA - peptide interactions involving chiral DNA mesophases: A geometric Analysis. J. Am. Chem. Soc 1996, 118, 6345–6349.